# Explore the Data Visualization Playground in Virtual Reality

Xuening Peng
Duke Kunshan University

Fateme Rajabiyazdi
Carleton Univerity

Xin Tong *
Duke Kunshan University

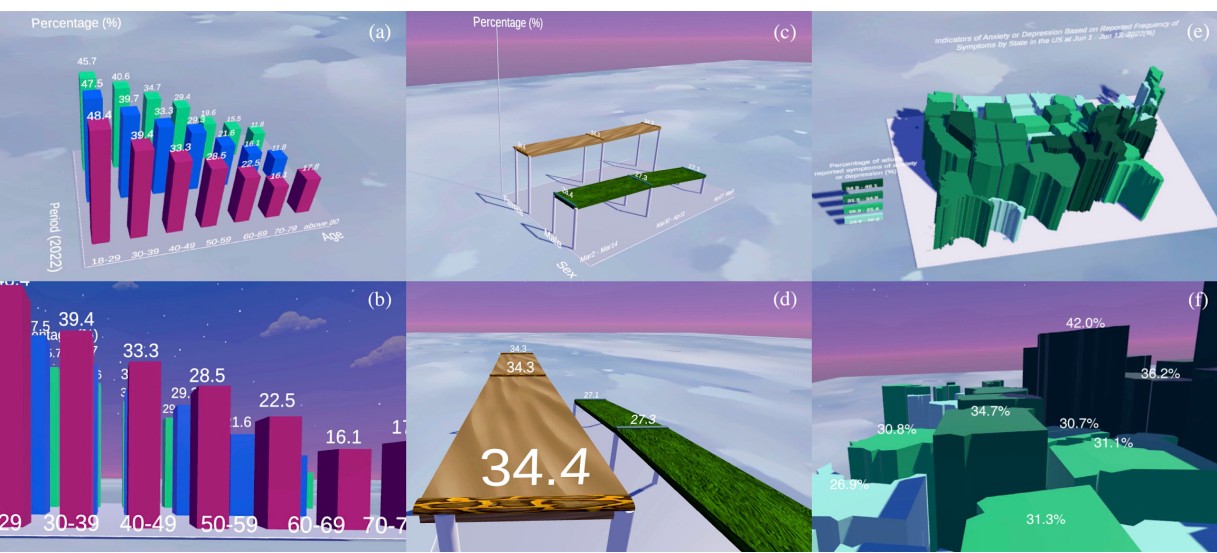

Figure 1: A collection of data visualization in VR (a) 3D bar chart (bird's-eye view) (b) 3D bar chart (close view) (c) 3D line chart (bird's-eye view) (d) 3D line chart (close view) (e) 3D Choropleth Map (bird's-eye view) (f) 3D Choropleth Map (close view). Refer to the short video that showcases our work in VR: *Video Demo*

## ABSTRACT

Data visualization in virtual reality (VR) shows great promises as a tool to represent complex and multidimensional datasets and provide immersive visual experiences. As there exist issues of visual blocks caused by scaling data objects and limited interaction settings in VR, in this research, we aim to design an autonomous and playful data visualization playground in VR by embedding interactive elements into the world. We transformed the commonly-used 2D visualizations, e.g., bar chart, line chart, and map, into 3D VR data visualization formats. Moreover, we also implemented interactions that allows users to walk, jump, slide, teleport, and switching viewpoints. We expect to provide a more clear and enjoyable data visualization representations for multivariate datasets through distinguishing colors, textures, and a customized interaction approach for each visualization. We believe our work can provide better insights to future design on how to construct an immersive VR world for data visualization. Next,we will conduct user studies to test the effects of VR interaction methods on people's potential perceptions and cognition.

## 1 INTRODUCTION

Data visualization is a graphical representation of data information, which has emerged as a powerful and widely applicable tool that enables humans to grasp trends, relationships, and patterns in a more accessible way [28]. More importantly, data visualization can enhance the clarity of the information and the aesthetic appeal of the data, through user interactions. Common visualization design idioms (visual representation form of the data ) include line chart, bar chart, scatter plot, pie chart, map, and so on. Conventionally, these 2D data visualizations remained in a static form, which appeared as fixed infographics and users can only look at limited information on a printed page [28]. However, in the era of Big Data, interactive data visualization idioms have been developed by researchers [8, 14, 21] to deal with large datasets and provide precise and accurate information. Visualization tools and platforms like ManyEyes [3], SAS Visual Analytics [5], Thoth [21], Tableau [6], and TOPCAT [7] make visualizations easier to interpret and produce. However, although 2D graphics are easy to implement, they may not work well to deal with multivariate data [18]. It is inevitable to label data with various colors, shapes, sizes, and interconnections when trying to display all multivariate data on one 2D screen. The giddy labels may cause great difficulties for the audience to comprehend the data, data information marked according to multiple criteria can be too complex to recognize [27].

Therefore, virtual reality (VR) has been identified as a promising medium that offers higher dimensionality for data representation and immersion experience for data viewers [18, 19, 27]. Previously, Bay-yari and Tudoreanu [9], DiBenigno et al [16], and Lee et al [22] have adopted VR as the medium to visualize their data. One remarkable advantage of VR is that it introduces a third Z-dimension into the space, where the data analysis can be easily and accurately applied to multidimensional and multivariate datasets without discarding details [17]. Scientific evidence showed that 2D visualization led to a huge degradation of quality compared to the capacity of the 3D environment as an effective platform for presenting complex data [20]. As for the immersion of the VR environment, users can take advantage of the greater space on offer, more natural interactions, and viscerally analyze 3D data through embodiment and immersion [15]. Researchers from Monash University and Microsoft explored "data

---
*Communication author: Xin Tong xt43@duke.edu
Data Science Research Center
Division of Arts and Humanities

visceralization" by providing one-to-one scale humanoids to show competition results of Olympic sprint and long-jump events [22]. They also prototyped famous architecture including the Eiffel Tower, CN Tower, and Burj Khalifa at a real-life scale to showcase and compare their heights of them to the views positioned from below. VR also acts as a problem-solving device, which transforms immense quantities of mind-breaking data into "graspable illusions" [12]. In particular, the Wall Street Journal [10] developed a VR guided tour of "21 years of the Nasdaq", illustrating the data fluctuations of Nasdaq's price or earnings ratio over 21 years by providing viewers with a stomach-dropping experience of taking a roller coaster. These data visualization projects in VR effectively immerse users into the data space by providing multidimensional visualization perspectives and vivid illusions in VR, successfully guiding users to naturally think, analyze, and digest data enjoyably.

However, there exists an essential limitation in data visualization in VR, which is the lack of interactions, especially autonomous and intuitive interactions with data. Firstly, visual blocks and perspective distortion have become a tricky problem in VR [26]. As data elements or objects tend to be designed much taller or on a one-to-one scale in VR to provide the users with a real-world experience [1, 22], they tend to cause some occlusion or blind spots because of visually overlapping positions and large size [19]. Secondly, the lack of autonomous and intuitive interaction design greatly reduces users' experience of immersion in VR. Literature [13, 23, 24] showed that autonomy plays an important role in raising customers' satisfaction and immersion level during tourism, which is similar to a data exploration tour in immersive VR. Nevertheless, most of the previous work either provided little interactions to avoid visual blocks or limited user autonomy to explore the datascape. The two data visualization projects "Manufacturing job prevalence in the US from 1939 to 2009" and "US electricity generation by source 1950-2020" developed by the "Flow Immersive" in VR downplayed the role of autonomous interactions, thus leading to less immersive experience [2]. They provide limited and simple interactions such as rotating and scaling the data frame but focus more on the visual effects of animation and color contrast. In addition, although the "21 years of the NASDAQ" VR tour [10] provides the interactive elements of switching viewpoints and jolting with data fluctuations, the viewpoints are only fixed to two positions and the moving routes are restricted within a single platform. Users would fail to explore Omni-bearing perspectives in the VR space and can only passively experience the moving pace designed in advance. Therefore, it is necessary to construct a fully autonomous VR environment to enhance users' immersive experience in data visualization. To achieve this goal, we are trying to add more interactive elements that allow a higher degree of autonomy for the users to have a full exploration of VR during the data visualization journey.

In this work, we aim to explore "how to visualize data in VR to create an immersive environment so that people could experience, understand and enjoy playing with the data?" We modeled a VR world with 3D bar charts, line charts, and maps with additional interaction settings (i.e., walk, jump, slide, teleport and switch viewpoints), expecting to create a more immersive, playful, and autonomous environment for users to interact with. Our work will contribute to filling the gap of previous work in VR by allowing a highly autonomous data visualization environment in VR. In future work, we will try to design more immersive and intuitive interaction elements or effects and implement user studies to conclude with scientific inference.

## 2 VISUALIZATION DESIGN IN VR

Our goal is to transform the conventional 2D data visualization into a VR environment where people could immerse into and explore the multidimensional aspects of their data. To demonstrate this transformation, we selected the three most commonly used visualization

graphs (bar chart, line chart, and map) as a starting point.

### 2.1 The Dataset

To design our visualizations, we used a dataset from the Household Pulse Survey collaboratively collected by the National Center for Health Statistics (NCHS) and the Census Bureau [4]. The survey was designed to evaluate the impact of the coronavirus pandemic on anxiety and depression amongst people in the U.S. from May 5, 2020, to July 11, 2022. The resulting data revealed "the percentage of adults who report symptoms of anxiety or depression that have been shown to be associated with diagnoses of generalized anxiety disorder or major depressive disorder in the U.S." by age, sex, gender identity, sexual orientation, race ethnicity, education, disability status, and U.S. states. In our visualization, we extracted three valid indicators age, sex, and U.S. states to prototype the visualization effects.

### 2.2 The VR Environment Implementation

The virtual world for data visualization was set and prototyped the visualization models by Unity 2020.1 on Windows Desktop. We employed the HTC-Vive headset with SteamVR Unity Plugin installed to implement the interaction functions and effects in the VR environment. All the materials applied to our visualization models were imported from either the default model library or the Unity Asset Store.

### 2.3 Visualizations and Interactions

#### 2.3.1 The VR Bar Chart

In the VR bar chart (Figure 1a and 1b), we showed the percentage of adults who reported symptoms of anxiety or depression by "Age" from Mar 2, 2022, to May 9, 2022. Visually, we transformed the flattened 2D bars into building-like cuboids. In the 3D coordinate system, the X-axis "Age" was divided into 7 age groups from 18 to over 80, the Y-axis "Period" represented the three time period when the participants reported their symptoms, and the Z-axis "Percentage" referred to the percentage of adults reported symptoms of anxiety or depression. The pink, blue, and green series of cuboids (Figure 1a and 1b) along the X-axis stand for the percentage of adults with symptoms of anxiety or depression across the three time period from Mar 2, 2022, to May 9, 2022, across Y-axis. The height of each cuboid was consistent with the real value in the dataset (48.4% = 48.4 meters), but with a 0.2 downscale for viewing convenience according to the pre-set height of the user. Values of percentage were shown accordingly above the cuboids. As for the interactions in VR, users can be equipped with the VR headset and two controllers and they are free to move around and step through the datascape by tapping around the touchpad, and jumping on top of the cuboids by pressing the controller trigger to teleport to the desired location. Multi-perspective viewpoints were also provided in our environmental setting. The height of the user was pre-set to be 1.7-meter tall. The user can switch viewpoints between the bird's-eye view and the close view by pressing down the central button on the touchpad. From the bird's-eye view of the bar chart (Figure 1a), the user is initially placed on the 4-meter tall transparent plane above the ground. The user can autonomously turn and move to any orientation, walking around, and looking down at the whole picture of these cuboids. Dimensional information (X, Y, Z axes) and general data trends can be easily grasped from a distant view. When pressing the central button of the touchpad or directly teleporting to the ground surface, the user comes to a close view of a taller, larger, and more visceral data world (Figure 1 (b)). By looking up, walking through, and jumping upon the cuboids, the user can actively explore the large-scale datascape and feel as immersive and visually impressive.

### 2.3.2 The VR Line Chart

In the VR line chart (Figure 1c and 1d), we presented the percentage of adults who reported symptoms of anxiety or depression by "Sex" from Mar 2, 2022, to May 9, 2022. To add a third dimension to a line chart, we transformed thin lines into bridge-like platforms supported by poles. In this 3Dcoordinate system, the X-axis "Period" represented the three time periods when the participants reported their symptoms, Y-axis "Sex" included male and female, and Z-axis "Percentage" referred to the percentage of adults who reported symptoms of anxiety or depression. We used two different materials from imported unity assets for each "bridge", wood and grass, to represent data collected from females and males respectively to not only distinguish the two groups, but wanted to provide a real-life experience of "crossing the bridge". To easily identify the turning point of the platforms, we also designed horizontal rods to suggest a period change along the X-axis and data labels displayed. The height of each pair of supporting poles was consistent with the one-to-one scaled value (34.4 % = 34.4 meters), and with a 0.2 downscale similar to the bar chart. The interaction modes of the VR line chart in VR involve teleporting, walking, and sliding. The user can teleport onto the bridge platform to walk over the woods or grass. It can also jump down to the ground level to move within the bridge field. Especially, by lifting the controllers, the user can slide on the bridge to experience the change of data across the three periods (though there is only a gentle slope in our case because of the feature of selected data). In terms of viewpoints, the bird's-eye view of the line chart (Figure 1 (c)) enables the user to look down at the X, Y, and Z axes and general data differences from a distant view. When transforming to the close viewpoint, the user is expected to climb up or slide down the slope of the platforms (Figure 1 (d)).

### 2.3.3 The VR Choropleth Map

The VR choropleth map (Figure 1e and 1f) displayed the distribution of the percentage of adults who reported symptoms of anxiety or depression across the national states in the US, from Jun 1, 2022, to Jun 13, 2022. The map shows the geographical information of the United States while the color and height visual channels represent the percentages of adults who reported symptoms of anxiety or depression. The color encoder gives the four levels of percentage values which are 24.5-29.8, 29.9-31.4, 31.5-34.8, and 34.9-46.1 to indicate a regional situation of the percentage of adults who reported symptoms of anxiety or depression in the US according to the original classification in the dataset. The darkest green represented the highest level of percentage (34.9-46.1) while the lightest green stood for the opposite (24.5-29.8). The stereoscopic legends of these percentage ranges were placed right beside the map. Besides, the height encoder represented the specific percentage value of each state, providing an accurate and intuitive view for the user. The area of the map was set to a decent scale for the user to move around and the heights of the regions were 20% downscaled by their original values for visual comfort for the 1.7m user. The interactions with the VR choropleth map allow its user to teleport to, jump upon, and walk through the stage-like regional zone from a close view (Figure 1 (f)). The users can also switch to the bird's-eye view to see the whole map from a top-down perspective (Figure 1 (g)). The double encoding (color and length) was set to ensure that the user can have a clear perception of the data whether viewing at close or distance. The region maps of different heights in close view can provide the user with a visual impression and perception of the differences in the percentage of adults who reported symptoms of anxiety or depression among states. On the other hand, the bird's-eye view makes the height difference of the region patches hard to detect while the user can gain information via the color encoder instead. The two encoders complement each other' s visual dead zone while switching the views.

## 3 DISCUSSION AND FUTURE WORK

We proposed various interaction approaches and s created an autonomous and playful VR environment for users to be immersed with their data. Users experience a higher degree of autonomy and flexibility through direct interacting with their data and switching viewpoints in VR. Rather than viewing a static position or being limited to a pre-set moving route, users h can explore the space following their own perceptual experience about the data by walking, sliding, jumping, or switching viewpoints. These basic interactions are natural and intuitive since they follow human behaviors in the real world, which could further increase users' immersion levels [25]. Our work fills the gap of occlusion or blind spots blocked by large-scale data objects in the VR space and sheds light on a future direction for constructing a "playful" and enjoyable VR data playground. We expect that these actionable interactions within immersive spaces inspire people's insights into the datasets. We believe that users will discover meaningful patterns, causations, and correlations in a playground environment while having fun, aligned to the fundamental goals of better understanding data in VR visualizations [11].

Future work will further explore the design of the VR data visualizations and interactions to give the users a feeling of being part of the data. For instance, we plan to implement and test more direct-manipulation interactions, allowing the users to manipulate data by lifting data blocks, and users can also rearrange some parts of data in the targeted dataset and compare their visual looks and data information. Other forms of interactions will also be examined, such as adding dynamic animations to the cuboids, similar to a moving pattern of elevators rising and falling. Additionally, we plan to allow users to adjust their own height in the VR environment to better evaluate the dataset and study the effect of one's perspective height on their perception of the visualizations. . We hypothesize that the dynamic interaction may enlarge users' sense of height or the length of the data pieces, thus altering their perception of the data. Our goal is to allow users to live as part of the data in the VR world, instead of being an inspector standing outside of it. Other components such as viewpoints, color selection, and scales of data will also be considered since they may have potential impacts on users' perception of data in the virtual world.

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
