# OpenReview forum: "Explore the Data Visualization Playground in Virtual Reality"
_IEEE.org/2022/Workshop/altVIS — Reject_

### Official Review · Reviewer_bieU · 2022-08-24

**Review:**

The research provides an innovative and eye-catching visualizing using virtual reality (VR).
pros:
1. the presentation of visualization in VR is innovative;
2. the types of visualization are diverse.

Cons:
1. the presentation needs to be improved. There are still many typos and grammar mistakes.
2. the discussion on relevant literature in VR visualization needs to be improved.


**Conflicts:**

The corresponding author Prof. Xin Tong is my co-author on other research.

**Review Inclusion:**

Yes

**Sufficiently Alt:**

Yes

**Superlative:**

Most eye-catching

---

### Official Review · Reviewer_vXHn · 2022-08-30

**Review:**

Summary review: The paper re-creates three common chart types (line, bar, and choropleth) in VR using a sample data set. The motivations and design choices are not well supported, and the paper feels incomplete. Additionally, this paper would be better suited, with significant improvement, for a main conference track or another workshop, and does not meet the "sufficiently weird" alt.VIS criteria. For example, there are specific sessions at VIS each year for VR papers that I suggest the authors review, e.g. https://virtual.ieeevis.org/year/2021/session_v-full-full3.html . I would therefore not endorse this paper to be accepted to alt.VIS.

Bibliography is incomplete/incoherent; for example, the line “virtual reality (VR) has been identified as a promising medium that offers higher dimensionality for data representation and immersion experience for data viewers” cites three sources, two of which are emails (only one of which specifies with whom the email took place), and the third specifically identifies a number of challenges with data visualization in VR. Other citations are mediocre sources at best, e.g. this blog post from LinkedIn: https://www.linkedin.com/pulse/5-reasons-use-virtual-reality-data-visualisation-jeremy-dalton The authors are encouraged in the future to consider literature from the visualization research community, such as proceedings of IEEE VIS, TVCG, ACM CHI, KDD, etc.

The paper repeatedly refers to “higher dimensionality” in data; but it is unclear how VR adds more than one dimension (from 2D to 3D). The paper doesn’t engage with existing methods for visualization of high-dimensional data such as embedding methods, matrix representation, small multiples, etc. It is unclear that the authors understand what is meant by the challenges of high dimensional data; for example:  “To add a third dimension to a line chart, we transformed thin lines into bridge-like platforms supported by poles.” — this isn’t adding a third dimension in the same sense that we refer to dimensions when we talk about “high-dimensional” data — this is just extruding a 2D line into a 3D plane.

The paper needs to be reviewed using a tool like Grammarly for syntactical problems.

There are a lot of subjective words used; for example: “These data visualization projects in VR effectively immerse users into the data space by providing multidimensional visualization perspectives and vivid illusions in VR, successfully guiding users to naturally think, analyze, and digest data enjoyably.” How do we evaluate what is “effective”, “successful”, “natural”, or “enjoyable”? This is unusual for an academic paper.

The paper does a nice job of addressing problems and challenges of data visualization in VR, and setting forth the paper’s objectives of improving user autonomy. However, the authors don’t elucidate the ways in which they have provided autonomy beyond what is already afforded inherently in an environment like Unity; furthermore, the authors introduce explicit problems in VR like occlusion which are then present and not addressed in their final visualizations.

The organization of the paper is difficult to follow; it is recommended to consider “related work” and “methods” as section headers, because the introduction spends a significant amount of time critiquing prior work, and the second section reads like a methods section.

The bulk of the paper (section 2) contains no citations to existing literature, which are warranted to justify design choices. For example, “The height of the user was pre-set to be 1.7-meter tall.” Why? “we extracted three valid indicators”: What makes an indicator valid?

There seems to be an attempt to make a playful metaphor here, but the metaphor is incomplete and therefore lost on the reader. For example, “We used two different materials from imported unity assets for each “bridge”, wood and grass, to represent data collected from females and males respectively to not only distinguish the two groups, but wanted to provide a real-life experience of “crossing the bridge”. Why is the metaphor a bridge? Why are these textures chosen? Is this an extension of the playground metaphor, or is this related to some other narrative metaphor (e.g. “we’ll cross that bridge when we come to it”?)

“The color encoder gives the four levels of percentage values which are 24.5-29.8, 29.9-31.4, 31.5-34.8, and 34.9-46.1” these values provide nothing for the paper narrative and are unnecessary. The data included in the paper doesn’t seem relevant and would best be left to an appendix or left out altogether.

**Conflicts:**

No conflicts.

**Review Inclusion:**

Yes

**Sufficiently Alt:**

No

---

### Official Review · Reviewer_geau · 2022-08-31

**Review:**

The authors of this submission propose to re-create three common visual representations in VR. The paper is quite interesting to read but I would argue that there are some concerns pertaining to its publication at alt.VIS. I will details those below.

First and most importantly, I do not think that the paper is a good fit for the workshop. The submission reads as a more classical paper that could be accepted at a workshop or symposium on virtual reality as its goal is to advance VR (VIS) research.

Second, the submission seems to be lacking some rigorous back up and review of the literature. Some of the references appear to be private email conversations, which, although they could sometimes be accepted in academic submission, would not make the cut here. The visualization and virtual reality literature has enough strong papers that could back the authors' claims that they would not need to rely on private exchanges.

Third, I fear that the submission currently does not consider the already tremendous work that has been done on visualization in VR. Notably, I would argue that the DXR [A] could be referenced by the authors and would be a good starting point to find more work done in VR or AR (through the citations of this paper).

As a minor point, the paper would surely benefit from some extra proof-reading. While it does not hinder the comprehension of the manuscript too much it would make it an easier read.


Overall, I would argue that the submission is primarily not a good fit for the alt.VIS workshop and I would thus argue that it should probably not be accepted as is.

[A] DXR: A toolkit for building immersive data visualizations


**Conflicts:**

No conflict of interest with the authoring team.

**Review Inclusion:**

Yes

**Sufficiently Alt:**

No

---

### Official Review · Reviewer_LDF5 · 2022-08-31

**Review:**

Meta Review:
In this paper, the authors use VR to visualise data using 3 types of visualisations (line, bar, chloropeth) and explore how to navigate inside them. Reviewers found this topic interesting but they agree that it has the following issues:
(1) The litterature on Immersive Analytics in general is not explored
(2) Some sources can be problematic as impossible to check, like e-mails
(3) The presentation (both the structure and the writing) could be improved for readability.

Overall, we decided to reject this paper for this year AltVis and encourage the authors to resubmit an improved version next year.

**Conflicts:**

No Conflict

**Review Inclusion:**

No

**Sufficiently Alt:**

No

---

### Decision · Program_Chairs · 2022-08-31

Reject